# What are the Main Sensor Methods for Quantifying Pesticides in Agricultural Activities? A Review

**DOI:** 10.3390/molecules24142659

**Published:** 2019-07-23

**Authors:** Roy Zamora-Sequeira, Ricardo Starbird-Pérez, Oscar Rojas-Carillo, Seiling Vargas-Villalobos

**Affiliations:** 1School of Chemistry, Costa Rica Institute of Technology, Cartago 30102, Costa Rica; 2School of Chemistry, Universidad Nacional, Heredia 86-3000, Costa Rica; 3Central American Institute for Studies on Toxic Substances (IRET), Universidad Nacional, Heredia 86-3000, Costa Rica

**Keywords:** pesticides, biosensor, sensor, transducers, agricultural production

## Abstract

In recent years, there has been an increase in pesticide use to improve crop production due to the growth of agricultural activities. Consequently, various pesticides have been present in the environment for an extended period of time. This review presents a general description of recent advances in the development of methods for the quantification of pesticides used in agricultural activities. Current advances focus on improving sensitivity and selectivity through the use of nanomaterials in both sensor assemblies and new biosensors. In this study, we summarize the electrochemical, optical, nano-colorimetric, piezoelectric, chemo-luminescent and fluorescent techniques related to the determination of agricultural pesticides. A brief description of each method and its applications, detection limit, purpose—which is to efficiently determine pesticides—cost and precision are considered. The main crops that are assessed in this study are bananas, although other fruits and vegetables contaminated with pesticides are also mentioned. While many studies have assessed biosensors for the determination of pesticides, the research in this area needs to be expanded to allow for a balance between agricultural activities and environmental protection.

## 1. Introduction

Since the beginning of the 20th century, due to commercial interests, extensive areas of tropical forests—mainly those in the Caribbean area—have been transformed into banana plantations. Banana producing regions have suffered significant environmental deterioration due to contamination by chemicals which have been detected at dangerous levels in soil, air and aquifers. In recent years, due to the increased in banana-growing activity, the use of pesticides has increased, generating a potential risk to health [1]. Of the vast number of pesticides used to grow bananas, many are difficult to determine and require sample preparation and conditions that make tests very laborious. Such control must often be carried out in situ and in real time, which would require new analytical techniques that can quickly provide answers and have low pesticide concentration detection limits. Pesticides used in agricultural applications have been associated with multiple health problems, such as cancer, sterility, deformations in fetuses, allergies, acute intoxications and even death [2]. Traditional analyses of environmentally persistent pesticides in the banana agroindustry include the following: high-performance liquid chromatography, capillary electrophoresis and mass spectrometry. Nevertheless, these methods have certain limitations, such as a high complexity, time consuming sample preparation, expensive instrumentation and requirement for highly skilled personnel [3]. However, several alternative methods allow for quantification in a shorter period of time and with a reasonable accuracy. For example, a promising field of research in this area is related to sensors and enzymatic biosensors that are used as suitable devices for rapid analysis [4]. In particular, the detection of organophosphate and carbamate pesticides, based on the principle of the inhibition of cholinesterase for pesticides [5] is an interesting method.

A sensor is defined as a device that obtains and responds to stimuli and signals originating in the environment. It is composed of two parts: A receiver and a transducer. A receiver receives the physical/chemical stimulus and transmutes this information into electrical energy, while a transducer allows for invaluable energy and an analytical signal that can be analyzed thoroughly and presented in an electronic form. On the other hand, biosensors are part of the recent classification of sensors provided by the International Union of Pure and Applied Chemistry (IUPAC). The differentiating characteristic of a biosensor is the presence of the biological/organic recognition (enzyme, antibody or nucleic acid) element that allows for the detection of particular molecules in the medium [6].

Biosensors have the potential to complement or replace classical analytical methods by simplifying or eliminating sample preparation. They have the potential to make field tests easier and faster, as well as decrease the cost per analysis.

In this review, we provide an overview of different classical and novel methods for the detection of pesticides in agricultural (fruit and water) samples. Novel techniques, focusing mainly on biosensors for pesticide detection, and efforts to improve their sensitivity and selectivity are presented in the following sections.

### Pesticides Use in Agroindustry Activity in Costa Rica

One of the main agricultural activity products in Latin American is banana fruit, which is produced for self-consumption or exportation. Not surprisingly, Latin American countries lead the global banana economy, not only because of their share of world trade, but also because of their higher capacity to respond to changing market conditions, when compared to other regions [7]. In Costa Rica, the banana agroindustry is an economic and social engine for the country, generating around 40,000 direct and 100,000 indirect jobs. The activity is concentrated in the Atlantic zone, where 76% of the local workforce is involved. Costa Rica is one of the three most important banana exporting countries in the world, exporting around 120 million boxes per year, which represents nearly $1 billion in annual foreign exchange. The country is one of the main global exporters of bananas. Of the total banana exports, 55% goes to Europe and 36% to the United States [8]. Meanwhile, the golden pineapple variety, produced in Costa Rica, predominates in the market as a global favorite. Europe is one of the main clients of Costa Rica. This continent consumes 40% of worldwide exports. Costa Rica is its main supplier in Central America, with 92% of exports. In Costa Rica, pineapple fields encompass an area of around 45,000 hectares, mainly in the north of the country [9].

Not surprisingly, the highest percentage by volume of imported pesticides in Costa Rica is used in the banana agroindustry, particularly fungicides and insecticides, which are intensively used to prevent Black Sigatoka and other diseases. Approximately 76 kg of active pesticide ingredients (i.e., fungicides, nematicides-insecticides and herbicides) are applied annually per hectare in the banana agroindustry [10]. Some quantities of the main pesticides used in Costa Rica are shown in Table 1. According to the US Environmental Protection Agency (EPA), pesticides are defined as any substance or mixture of substances intended to prevent, destroy, repel or mitigate pests [11]. These substances can be classified based on their target organism, origin and chemical structure. These can be inorganic, synthetic or biological (biopesticide) compounds [12], including substances intended to be used as regulators of plant growth, defoliants, desiccants, fruit thinning agents (or substances used to prevent premature fruit loss), and substances applied to crops, before or after harvest, to protect deterioration during storage and transportation [13].

There are different ways to classify the great variety of pesticides, either considering their targeted pest, their mobility in plants, their toxicity, the fate of their application, their chemical structure or their origin. Mainly, individual products are classified in a series of tables, according to the oral or dermal toxicity of the technical product. The tables are subject to periodic review [14]. Considering their target, pesticides are classified into insecticides, acaricides, bactericides, herbicides, fungicides, rodenticides, nematicides, molluscicides, avicides and algaecides. Because of their mobility, pesticides can be applied systematically or through contact, in which case, they are applied to one part of the plant and through the foliage, so that they can reach other nontreated parts. Considering their toxicity, the World Health Organization (WHO) groups pesticides into four classes: IA (extremely toxic), IB (highly toxic), II (moderately toxic), and III (slightly toxic). This classification is adopted by most countries in Latin America. Depending on the destination of their application, these classifications can be used for sanitation, livestock, domestic activities, personal hygiene or in the food industry. Pesticides are grouped by their chemical structure into carbamates, chlorinated substances, phosphorous substances, inorganic compounds, pyrethroids, thiocarbamates, urea derivatives, arsenicals, bipyridyls and many others [15].

In this context, the intensive use of pesticides, the extensive drainage of banana plantation systems and the high volume of precipitation increase the potential toxicity of these molecules in aquatic ecosystems and also in the human population [16].

A maximum residue level (MRL), fixed by The European Commission, corresponds to the highest level of pesticide residue that is legally tolerated, in or on food or feed, when pesticides are applied correctly (Good Agricultural Practice). The MRLs for all crops and all pesticides can be found in the MRL database of the Commission, for example, in the case of beet, the MTR value for chlorothalonil is 0.01 mg/kg. In particular, this pesticide and others were detected above the allowed limits in some agricultural regions of Costa Rica [8]. Concentrations of 0.06 mg/kg in beet, 0.12 mg/kg in cabbage, 0.3 mg/kg in broccoli and 0.2 mg/kg carrot have been reported and are clearly values that exceed the MTR limits [17]. Moreover, in soil and surface water, average concentrations between 0.24 mg/kg and 0.70 ug/L [17] were also determined. According to the Maximum Tolerable Risk (MTR), the surface water for ethoprophos and chlorpyrifos is on the order of 0.063 and 0.003 µg/L. High pesticide residues were detected in surface water samples, in the surroundings of the banana plantations of Costa Rica, in higher levels than those allowed for ethoprophos and chlorpyrifos, i.e., between 1.5 and 0.7 µg/L, respectively [10].

Arias-Andres et al. conducted a 4-year risk assessment at the Madre de Dios River (RMD) and the coastal lagoon on the Caribbean coast of Costa Rica, close to banana and pineapple crops. They analyzed the species sensitivity distributions (SSDs) in order to derive hazardous concentration HC_5_ values for the pesticides. The hazardous concentrations for 5% of the species, diuron and ametryn herbicides, carbofuran, and diazinon and ethoprophos insecticides, exceeded the HC_5_ value or the lower limit of their 90% confidence interval, suggesting that they were above the accepted levels and exposed to toxicity risks [18].

## 2. Recent Progress in Pesticide Determination

Currently, there are several methods to determine agricultural pesticides in fruits and in the surrounding environment, such as surface waters and soil. In Costa Rica, the most frequently encountered pesticides are fungicides such as thiabendazole, propiconazole and imazalil; nematicides, such as terbufos and cadusafos; and the insecticide chlorpyrifos [19]. Among the classical methods, used for the analysis of pesticides in water samples, gas chromatography (GC) and liquid chromatography (LC) are the most commonly used [20]. A comparison between techniques is summarized in the Table 2. For example, chromatography-based methods have been the most widely used techniques for the analysis of organophosphorus pesticides (OOPs). Many researchers around the world have adopted different extraction procedures for the estimation of OPPs using chromatography [21,22]. However, the low detection levels for these compounds in environmental matrices indicate that the use of new analytical instruments with a high selectivity and detectability, such as gas chromatography-mass spectrometry (GC-MS), liquid chromatography-tandem mass spectrometry (LC-MS/MS), and ultra-high-performance liquid chromatography-tandem mass spectrometry (UHPLC-MS/MS), is often necessary to determine these pesticides [23,24,25]

New specific pesticides cannot be detected using GC-MS due to their thermal instabilities and polarities. A practical technique that solves this difficulty is liquid chromatography−mass spectrometry (LCMS), which has been widely used to quantify LC-amenable (or thermally labile) pesticides and to confirm their identities in vegetables and fruits at low parts-per-billion (ppb) concentration levels. Alternatively, an LC can be combined with full scan mass spectrometers, such as Orbitrap and time-of-flight (TOF) mass spectrometers. These types of MS have increasingly been used for quantification, identification, characterization, and the structural elucidation of pharmaceuticals, pesticides, veterinary drugs, unknown contaminants, excellent full MS scan sensitivity, and complete mass spectral information [26]. In recent times, the development of miniaturized chromatographic methods has attracted the interest of many researchers, providing a higher sensitivity than that obtained with conventional HPLC. Capillary-/nano-liquid chromatography (CLC/nano-LC) offers the possibility to perform rapid, highly efficient analyses in different fields. The sensitivity increases, using LOD and LOQ, within the 4.4–37.5 and 14.5–125.0 ng/mL ranges, respectively. This method is an alternative for detecting pesticides [27].

Proficiency tests by inter-laboratory comparison are a successful tool to improve the control of the analytical laboratories [28]. In addition, it is a requirement to ensure the veracity of the data obtained to opt for the accreditation of tests under the ISO/IEC 17025: 2017 (technical committee of the International Organization for Standardization (ISO) and the International Electrotechnical Commission (IEC)). A parameter to evaluate the laboratory results is the z-score (z) (the signed fractional number of standard deviations by which the value of an observation or data point is above the mean value of what is being observed or measured), which is set in ISO/IEC 17043, and a |z| ≤ 2.0 is considered acceptable [29]. In recent years, there has been a significant increase in these types of inter-laboratory studies of the pesticides in fruit and vegetables, with more than 30,600 pesticide residue results using multi-residue methods [29].

The development of sensor and biosensor devices is a novel strategy, and these devices have several advantages over traditional methods, for example, in their simplicity, sensitivity, selectivity and capacity to be deployed in the field, which is still highly desirable for the monitoring of pesticide contamination [30]. Moreover, the introduction of nanomaterials in the construction of sensors and biosensors emerges as a highly efficient analytical tool for the detection of pesticides and other analites. It stands in contrast to conventional techniques, which have various complications, including sample decomposition, a limited number of samples in a specific time interval, matrix interference and a longer test duration [3,31]. In the detection of pesticides, biosensors pave the way to a more efficient analysis, with greater precision, lower detection capabilities, continuous monitoring and a low cost [32]. In Table 3, different types of sensors, modified with several nanomaterials, for the detection of pesticides are summarized. This table compares the pesticide type, limit of detection, generalized practical applications and future challenges.

### 2.1. Electrochemical Sensors Based on Nanomaterials

The electrochemical method for pesticide analysis depends on the selected process, its optimization, and the appropriate working electrode. The method occurs when the reaction of interest occurs, and the working electrode acts as a transducer. The electrode material depends on the possible redox of the analyte and the working potential [88].

Several examples have been discussed in the literature. For example, the use of a nanomaterial-modified electrode, based on the polymer, poly(3,4-ethylenedioxythiophene) (PEDOT), and carbon nanotubes (CNT) to determine the presence of carbamates (mancozeb, (MCZ)) in a water sample was described by Zamora and co-workers, using cyclic voltammetry (CV) as an electroanalytical method [89]. In cyclic voltammetry, the modified electrode exhibited electrocatalytic activity towards the oxidation of MCZ, with a working linear range of 25–150 µmol/L and a limit of detection of 10 µmol/L. The developed electrochemical sensor provides economic and environmental advantages. The results aimed to use a new electrochemical method for the in situ and real-time detection of pollutants.

In a similar approach, Della Pelle et al. developed a carbon nanosphere (CB) of the screen-printed electrode (SPE) for the detection of four carbamates (i.e., carbaryl, carbofuran, isoprocarb and fenobucarb) carbofuran (CBF) in samples of corn and wheat. The device showed a very stable and reproducible response, with low work potentials, an excellent resistance to contaminants and a nanomolar range limit of detection. Further, the device was validated with a UHPLC-MS/MS procedure, confirming the accuracy of the method [44].

Gold nanoparticle electrodes, based on the molecular polymer printing method (MIP), reduced with graphene oxide and coated with citrate (AuNPs)/(3-mercaptopropyl) trimethoxysilane (MPS), were studied by Tan and Song et al. [47,48] Both systems reported an LOD of 2.0 × 10^−8^ mol/L and 1.0 nmol/L (20 nmol/L and 1.0 nmol/L, with a linear range from 5.0 × 10^−8^ to 2.0 × 10^−5^ mol/L and from 0.003 to 2.00 μM, respectively). The electrodes showed a high adsorption capacity and excellent selectivity for CBF in real samples of vegetables and fruits. [47,48] Additionally, the derived activated carbon (PET-AC), prepared from waste bottles of polyethylene terephthalate (PET), was used to modify the glassy carbon electrode (GCE) (PET-AC/GCE) for the detection of CBF. The electrochemical activity was measured by means of CV, showing an LOD of 0.03 μM and a remarkable sensitivity of 0.11 μA μM^−1^ cm^−2^. The PET-AC/GCE maintains a high selectivity towards potentially interfering species, as well as a high stability and repeatability [45].

Chlorpyrifos (O,O-diethyl-O-3,5, 6-trichloro-2-pyridylphosphorothioate) is an organophosphate insecticide used in agriculture [90]. Some approaches have been suggested for its detection in field crops, such as fruits, vegetables, and cotton. An electrochemical nanosensor, based on tin oxide with fluorine (FTO) and decorated with gold nanoparticles (AuNPs), was reported by Tallan and coworkers [50].

Under optimized conditions, the proposed FTO-based nanosensor showed a high sensitivity, from 1 fM to 1 μM, and a stable response, with an LOD up to 10 fM, in the detection of ChP. The FTO-AuNPs sensor was used successfully for the detection of standard ChP and also in real samples of up to 10 nM for apple and cabbage and 50 nM for pomegranate. The proposed FTO-AuNPs nanosensor can be used as a quantitative tool for the rapid on-site detection of ChP traces in real, miniaturized samples due to its excellent stability, sensitivity, and simplicity. 

The fungicide dimethyldithiocarbamate (DMC) and the herbicide diuron (DU) were analyzed in an aqueous solution using the square wave voltammetry technique, with an LOD of 0.02 mg/kg and 2.6 × 10^−7^ mol L^−1^, respectively [51,52]. The DU was also analyzed using a microbial biosensor, based on the cyanobacterium *Anabaena variables*, for the detection of on-line herbicides by inhibiting the generated photoelectric current. The biosensor with bacterial cells was immobilized on a carbon felt electrode, using alginate as a trapping polymer and benzoquinone (BQ) as a redox mediator to support the electron transfer [53].

Glyphosate (GF) is a herbicide that is widely used in crops. It has been determined by electrochemical sensors, with a good accuracy in several water samples, using amperometric and potentiometric methods. Khenifi et al. developed an amperometric sensor, based on an electrode of Ni_1_-xAlx (OH) 2NO_3_x ∙ nH_2_O-layered double hydroxide (LDH). The NiAL-LDH-modified electrode, prepared by coprecipitation or by electrodeposition on the Pt electrode surface, shows an LOD of 0.9 mM. The authors suggest that their electro-detection is based on the oxidation of the amine group, in their structure, by Ni(III) centers [57]. A similar approach was taken by Vaghela *et al*., using a biosensor based on guar agarose gum (A-G), trapped by urease bio-nanoconjugate with gold nanoparticles (AuNps). The biosensor is based on the inhibition of urease activity by glyphosate, which is measured with direct potentiometry using a selective ammonium ion electrode, covered with a urease nanoconjugate membrane, A-G. The composite biopolymer used for immobilization produces thin, transparent and flexible membranes with a superior strength and mechanical stability. The conjugation of urease with AuNps improves the response characteristics for potentiometric measurements. The biosensor shows a linear response at the glyphosate concentration of 0.5 to 50 ppm, with a detection limit of 0.5 ppm [58].

Recently, magnetic particles modified with antibodies were reported in the detection of glyphosate in beer. The calibration curve ranged, in terms of concentration, from 0 to 10,000 ng/L, with a detection limit of 5 ng/L and a limit of quantification of 30 ng/L. The assay was demonstrated to be cost-effective in comparison to a colorimetric ELISA [54].

Other approaches using nanomaterials, like carbon nanotubes (MWCNTS)/copper oxide nanoparticles (CuO), have been reported [4,57,91]. Some limitations of electrodes for electrochemistry are cleanliness, stability, and reusability, and some products can adsorb onto the electrode surface, causing electrode pollution and a reduction of the measured current, affecting the selectivity. [92] Electrodes used up to 50 times in measurements of pesticides, with a reduction of 25%, compared to their first reading, were reported. [93] To increase the reusability of the electrochemical sensors, nanomaterials that can retain their catalytic activity under complex biological systems and under varying physiological conditions can be used. In addition, a nanomaterial surface can be used in a controllable way for the measurements of multicomponent systems [94].

### 2.2. Optical Sensors Based on Nanomaterials

Optical detection for pesticide determination represents the highest percentage of flow methods designed. Spectrophotometry and luminescence are the most frequently employed. In particular, fluorescence is the most commonly used luminescence technique due to its higher selectivity and sensitivity, when compared to spectrophotometry. Optical flow sensors are based on the implementation of solid phase spectroscopy (SPS) in flow analysis. They are usually named flow-through optosensors or flow optosensors (FOs) [95].

Surface plasmon resonance (SPR) technology is an optical detection platform that offers a real-time and label-free analysis of molecular interaction. SPR-based immunosensors have been widely applied in the detection of large molecules, where the analyte mass and the use of the sandwich immunoassay format can lead to a high signal and thus the requirement for sensitivity [96]. Biosensors that interact with nucleic acid are called genosensors. In this type of interaction, a complementary base-pairing principle is used for recognition, for example, in DNA, adenine-thymine and cytosine-guanine [97]. The colorimetric sensors are efficient and simple. They have been used with gold and silver nanoparticles in the detection of dithiocarbamates, with an abrupt change of color that is visible to the naked eye [69].

In recent times, novel analytical methods have been developed for the detection of pesticides in fruits and vegetables, for example, surface-enhanced Raman spectroscopy (SERS), with nanomaterials, in which probed molecules are absorbed onto the roughened surface of transition metals, resulting in a significant enhancement of the Raman signals by many orders of magnitude in highly localized optical fields of nanostructures. Coupled with metal nanosubstrates, SERS can increase the intensity of Raman signals by more than one million times due to the effects of electromagnetic field enhancement and chemical enhancement [98].

The CBF is a common nematicide in agricultural practices; it is used with methods of optical transducers. An example of these transducers is the use of an immunosensor based on a microcantilever that quantitatively detects carbofuran, using monoclonal antibodies, against carbofuran, as receptor molecules, obtaining an LOD of 0.1 ng/mL in vegetable samples [59].

Three different methods of analysis of optical transducers were found for DU. One was based on MIP, with an LOD of 0.013 μg/mL [61]. An optical bioassay, based on green photosynthetic microalgae, is a promising alternative for DU monitoring in seawater, with an LOD of 0.3 nM [62] and a test biosensor based on photosystem II, with quantification limits of 1 nM. Masojidek et al. highlighted the ability of the biosensor to complete the analysis efficiently and quickly [63].

Additionally, an optical biosensor is used to measure the GF, in real time, of the cytotoxicity of xenobiotics built, in which the technology records integrated cellular responses, based on changes in cell morphology and dynamic mass. This biosensor has an LOD of 2.98 nmol/L and is used for samples of agricultural products [66].

The fungicides imazil (IZ) and thiabendazole (THB), were analyzed in a biosensor based on white light reflectance spectroscopy, with an LOD of pesticide concentrations as low as 0.6 ng/mL and 0.8 ng/mL in samples of water and grape wine, respectively. The recovery values ranged between 86 and 116%, showing the precision of the measurements made with this developed biosensor [60].

The herbicide tebuthiuron (TB), was analyzed using MIPs, with an LOD of 0.023 μg/mL in vegetables and a linearity coefficient (R) of 0.999926. Thiazophos (THP), an insecticide used in fruits, was analyzed, employing a non-competitive immunoassay for the detection of traces using a direct surface plasmon resonance (SPR) biosensor. Two anti-triazophos monoclonal antibodies (mAbs) were immobilized on the sensor chip and characterized using SPR-based kinetic analysis. The biosensor assay showed a high specificity and low detection limit of 0.096 ng/mL at THP, with a linear detection range of 0.98–8.29 ng/mL^−1^ [67].

#### 2.2.1. Chemiluminescence Sensors

Chemiluminescence of the surfaces of the electrodes is where the electron transfer reactions occur, forming excited states of light emission. In some cases, nano-materials are used to improve the sensitivity of some pesticides [83].

There are four reported detection methods for CBF, ChP, CBY pesticides in an aqueous solution. For the CBF nematicide, a set of chemiluminescent (CL) sensors, based on very simple and easy-to-use nanoparticles, for the discrimination of carbamate pesticides was developed. This CL sensor array is based on the simultaneous use of the triple-channel properties of the silver nanoparticle, functionalized with luminol (Lum-AgNP) and the H_2_O_2_-CL system, with an LOD of 24 μg/mL [80]. With this same nematicide, a method for detecting it in plants was found, based on an electrogenerated chemiluminescence (ECL) detection platform for the detection of CBF, based on the ECL energy transfer (ECRET) and aptamers marked with points of carbon (point C), as a recognition element. Gold nanoparticles loaded with fullerene (C_60_) (C_60_-Au), modified in a vitreous carbon electrode, were used as an energy donor. This method has an LOD of 8.8 10 × 13 mol L^−1^. This same method was developed for the insecticide, ChP, and the nematicide, CBY, for the aqueous samples, with an LOD of 24 μg/mL [79].

The strategy of developing a late-sensitive electrochemiluminescence (ECL) biosensor for the detection of glyphosate, based on assisted in situ generations over ZnS quantum dots (QD) in the mesoporous carbons substrate (OMC), was proposed. This ECL biosensor (QD), used to determine the herbicide, showed a broad linear range of 0.1 nmol/L to 10 mmol/L, with an excellent sensitivity, reproducibility and selectivity. This method had an LOD of 8929 nmol/L for vegetable juice samples [81].

The detection of ChP was carried out using two detection methods. The first method was based on graphitic carbon nitrite/bismuth ferrite nanocomposites (g-C_3_N_4_/BiFeO_3_ NC), synthesized by an easy sol-gel combustion method and used as a catalyst, similar to peroxidase. On the basis of the catalytic activity in the luminol-H_2_O_2_ reaction, the nanocomposites were used as a dual-reading colorimetric-chemiluminescent immunochromatographic assay (ICA) for the multiplexed detection of pesticide residues, using chlorpyrifos as an analytical model, with an LOD of 0.033 ng/mL for the aqueous samples. The advantages of this method include its low cost, time efficiency, high sensitivity and excellent portability [82].

A novel and highly sensitive electrochemiluminescence biosensor (ECL) system was designed and developed for OP pesticides, in which bimetallic Pt-Au nanoparticles are electrodeposited in multi-walled carbon nanotubes (MWNTs), with a modified glass carbon electrode (GCE), to increase the surface area of the electrode and the ECL signals of the luminol. Based on the effects of the dual amplification of nanoparticles and H_2_O_2_, produced by the enzymatic reactions, the proposed biosensor exhibits a high sensitivity, with concentrations in the range of 0.1 to 50 nmol/L for ChP, with a limit of detection of 0.08 nmol/L in cabbage samples [83].

#### 2.2.2. Fluorescence Sensors

The fluorescent method is based on increased/decreased emissions, when a fluorescent material is subjected to a variety of factors that are principally related to the change of an analyte concentration and to the chemical environment in complex solutions [84].

A ratiometric nanometer, based on quantum carbon dots (CQD), showed that dual-emission fluorescence was developed. The spherical fluorescent carbon nanoparticles were obtained from lactose using a hydrothermal method, for the nematicide CBF, with an LOD of 12.2 μmol/L for a sample of soy sauce [84].

The fungicide, dimethyldithiocarbamate (DMCM), is an assay based on the interaction of pesticide with AuNPs. In this process, zinc ions are released, and the fungicide adsorption induces the aggregation of the AuNPs. The aggregated AuPNs decrease the fluorescence intensity of the quantum dots (QD), CdSe/ZnS, covered with 3-mercaptopropionic acid through an internal filter effect. This method has an LOD of 2 nmol/L for both tomato and rice samples [85]. For the fungicide difenoconazole, a quantum dot (QD) method was developed with MIP for samples of apple, orange and fruit juices, presenting an LOD of 0.5 ng/mL [86]

There is a critical need for the development of rapid tools, in situ and in real time, to monitor the impact of pesticide discharge toxicity on aquatic ecosystems. For the DU herbicide, the chlorophyll A and fluorescence of the microalgae was evaluated. They used three populations of microalgae to develop the biosensor: *Chlorella Vulgaris*, *Pseudokirchneriella subcapitata* and *Chlamydomonas reinhardtii*. This method considered the parameters of the viability and sensitivity of the biosensor, such as the concentration of algae and the intensity of light, and allowed for the development of a novel, easy-to-use, low-cost and portable algae biosensor, with an LOD of 1 mg/L [87].

Similar to the GF method, the detection method based on magnetic fluorescence nanoparticles (FMP) was described by the quantitative fluorescence intensity and the specific application based on DNA. A sandwich format consisted of a DNA/FMP-conjugated glyphosate probe/target. The results were measured as a function of the fluorescence intensity, which was obtained by comparing the free glyphosates in concentrations of 1 to 10,000 nmol/L to detect the binding of the glyphosate-double DNA-FMP nanoparticles. Moreover, the quantitative information on the free glyphosate analysis was translated into a single-probe DNA signal, with an excellent linear correlation in the concentration range of 1–10,000 nM (R^2^ = 0.98) and a detection limit of 0.27 nmol/L [65].

A sandwich-type immunosensor composed of AuNPs, coated with target DNA using a double-antigen probe, was developed for the fluorescence intensity measurements and quantitative single-strand DNA analyses based on the concentration of free glyphosate. These intensities were obtained from a comparison with free antigens in levels of 0.01–100 μg/L for the detection of the antigen, immobilized with the biosensor [64].

#### 2.2.3. Colorimetric Sensors

In 2019, a rapid, manageable and straightforward advance in the field of nanoparticles, for the detection of pesticides in the environment and samples of juice, was demonstrated. That study uses copper nanoparticles, coated with cetyltrimethylammonium bromide (CTAB), as a colorimetric probe, with an LOD of 97.9 ng/mL [69]. This author also conducted a previous study with silver nanoparticles, covered with sodium dodecyl sulfate (SDS-AgNPs), with an LOD of 9.1 ng/mL in water [68]. The same method could detect the ChP insecticide in samples of water and rice, as well as in apple juice and green tea. These colorimetric biosensor methods use AuNPs of 13 nm, covered with citrate, for the detection and discrimination of several organophosphorus pesticides, with an LOD of 118 ng/mL [70]. This method is followed by a colorimetric biosensor that comprises five economic and commercially available indicators, sensitive to thiocholine and H_2_O_2_, targeting the activity of acetylcholinesterase (AChE) and developed with an LOD of 4.6 × 10^−8^ g/L. This same method also investigates the nematicide, Carbaryl CBY, with an LOD of 2.3 × 10^−8^ g/L [71].

### 2.3. Piezoelectric Sensors Based on Nanomaterials

For water samples, several studies developed a biosensor with a quartz crystal microbalance immunosensor (QCM) for pesticides, such as CBY, CBF, ChP, parathion (PHT) and GF, with LODs of 1 × 10^−7^ M and 1 mg/L for CBY [72,73]. Furthermore, the LOD for the following chemicals is as follows: CBF: 1.30 × 10^−9^ mol/L [99]; ChP: 1 × 10^−10^ mol/L & 250 μg/L [24,78]; and PHT, GF: from 4 μg/L [78] to 250 μg/L [24].

In the same way, using the QCM technique, CBY was detected in fruit and vegetable samples, based on two methods: the first method used monoclonal antibodies (MAb), and the second was based on a bulk acoustic resonator film. The former has an LOD of 11μ g/L, while the latter holds an LOD of 2 × 10^−10^ M [74,75]. For ChP, the technique of QCM was also developed with an immunosensor, with MIPs and an LOD of 250 μg/L. These polymers can provide a high degree of sensitivity and selectivity, while maintaining excellent thermal and mechanical stability [24].

### 2.4. Biosensors Based on Acetylcholinesterase Inhibition by Different Pesticides

A biosensor based on the inhibition of cholinesterase activity is one of the most widely used in the determination of organophosphate pesticides, such as chlorpyrifos. The analytical device that incorporates the enzyme is integrated into a physical-chemical signaling transducer or the transduction microsystem [3]. Then, the signal (electrode, optical detector, piezoelectric crystal, etc.) converts the biochemical response into amplified electric and optical signals, which are measured and decoded by an appropriate electronic unit. Using this technique, the analyte selectively inhibits the activity of the immobilized enzyme, resulting in a decrease in the signal that is proportional to the amount of target analyte present in the solution [100].

A sensitive electrochemical biosensor for carbofuran (CBF), based on acetylcholinesterase (AChE), with a modified vitreous carbon electrode, a polyaniline shell (PANI) and multi-walled carbon nanotubes (MWCNT), was assembled by Martinez and coworkers. The GC/MWCNT/PANI/AChE biosensor exhibited detection limits of 1.4 mol/L for CBF and was applied successfully in samples of cabbage, broccoli and apple. The results were validated through HPLC [4].

As for the amperometric biosensors, Montes et al. described the characterization and optimization of graphite-epoxy-AChE biosensors, which improve the required electrochemical properties, such as a high electron transfer rate, a high signal-to-noise ratio, and adequate sensitivity. The analyzed water samples exhibited an LOD of 0.00025 mg/L for the CBF [46].

The insecticide chlorpyrifos (ChP) was analyzed using an electrode of self-assembled monolayers (SAM) of single-walled carbon nanotubes (SWCNT), wrapped with a thiol-terminated single-stranded oligonucleotide (ssDNA) in gold, in a polyaniline matrix for the immobilization of the enzyme, acetylcholinesterase. This biosensor has a crucial step, in which a small pH change occurs in the vicinity of the electrode surface, employing the enzymatic reaction of AChE. The dynamic range to determine ChP is approximately 1.0 × 10^−6^ mol/L, with a good reproducibility and stability. The detection limit of the biosensor was 1 × 10^−12^ mol/L in the water samples.

The immobilization of AChE in modified nanocomposites has been seen as a powerful tool to increase the response of the biosensor in the detection of pesticides in fruit and vegetable samples. A ratio of 1:3 of tin dioxide (SnO_2_) and MWCNT nanoparticles was obtained by dispersing them in a 0.2% solution of CHIT. Then, a suspension of 2.5 μL of the compound, MWCNT-SnO_2_-CHIT, was used to modify the surface of a gold electrode. Another technique is to immobilize the enzyme in SPE through bioencapsulation in a gel, with an LOD of 0.05 μg/L [101].

For ChP, Guo et al. developed a portable waste detection instrument by integrating an amperometric acetylcholinesterase biosensor (AChE) and a signal detector. The AChE biosensor was composed of modified tin oxide nanoparticles (SnO_2_), chitosan and a nanocomposite of multi-wall carbon nanotubes. This instrument can perform a rapid detection of pesticide residues in fruits and vegetables at the site, with automatic data processing, visualization and data storage. The limit of detection was 100 ng/L. The measurement time from the sample treatment to exposure was 15 minutes. Compared with traditional analytical methods, this proposed pesticide residue detection instrument had a good precision and high stability [102].

Likewise, a piezoelectric biosensor, made of macromolecular polymers and carbon nanotubes of multiple carboxyl walls (MWNTs-COOH), coated with AChE, was used on the glass surfaces, coated with Ag. These biosensors were used to determine the pesticide residue, ChP, in freshly collected radishes, with an LOD of 51.40 ng/L [103]. For the detection of CBF in grain samples, the cost-effective QCM technique, with an acoustic micro-immunosensor and an LOD of 4.5 × 10^−6^ mol/L, was used.

Recently, the fabrication of stable and sensitive biosensors for ChP detection, based on the immobilization of AChE on the electrode surface of a boron-doped diamond (BDD) electrode, modified with a nanocomposite, prepared from carbon spheres and gold nanoparticles, was presented by Wei and co-workers. On the one hand, gold nanoparticles can provide a large surface area, improving the loading efficiency and electron transfer speed, while, on the other hand, carbon nanospheres may improve the structural and electromechanical properties and provide a good biocompatibility. The results demonstrate that the fabricated system exhibited a higher sensitivity, lower detection limit, good reproducibility and acceptable stability in organophosphate molecules detection [104].

## 3. Discussion

The use of pesticides has increased in the area of agriculture in recent years. The pollution derived from this activity affects water, vegetables, fruits, and even soils due to the waste generated by them. The toxicity of organophosphorus pesticides and the family of carbamates lies in the inhibition of acetylcholinesterase (AChE), which is of vital importance for the nervous system of humans and insects [5]. To protect human health from possible hazards, it is pertinent to develop sensitive, rapid and reliable methods for the determination of pesticides in water, vegetables, and fruits. Based on this, the detection limits of nine pesticides were compared: carbofuran, chlorpyrifos, malathion, methyl parathion, carbaryl, diazinon, diuron, glyphosphate and imazil. These were found using the methods based on the acetylcholinesterase inhibitor (Table 4), classical methods, such as gas chromatography and HPLC (Table 2), as well as methods based on sensors and biosensors. The pesticides were also tested in different matrices, including water, vegetables, fruits.

First, the aqueous matrices were analyzed. It was found that the lower detection limits for the CBF of the family of the carbamates are 2.87 × 10^−14^ μg/mL and 6.25 × 10^−7^ μg/mL, which correspond to a biosensor based on a piezoelectric method, with a transducer based on a quartz crystal microbalance immunosensor (QCM) [99], and an amorphous metal boron-based acetylcholinesterase biosensor [106], respectively. When compared to a HPLC-DAD-based method [23], which has an LOD of 2.00 × 10^−5^ μg/mL, it is evident that this is less sensitive than the biosensor-based methods.

The insecticide, ChF, had lowest values of LOD, at 3.50 × 10^−7^ μg/mL and 9.91 × 10^−7^ μg/mL, corresponding to a biosensor, with a SAM electrode of SWCNT, wrapped with oligonucleotide ssDNA in gold, in a polyaniline matrix for the immobilization of the enzyme, AChE [49], and the amorphous metal boride-based biosensor, respectively [106].

The classical analysis methods projected values of 1.00 × 10^−5^ μg/mL by means of a solid phase extraction with HPLC-MS/MS [35]. In a more specific water sample, taken from rivers, we obtained, as minimum values of the limits of quantification, 3.5 × 10^−14^ μg/mL and 2.06 × 10^−9^ μg/mL for the biosensor based on QCM [77] and an AChE/AuNPs/VNSWCNTs biosensor [107]. Meanwhile, the quantification limits for gas chromatography methods correspond to 6.00 × 10^−3^ μg/mL [42].

The pesticides, malathion (MTH) and methyl parathion (MPHT), show the following lower detection values of 2.78 × 10^−11^ μg/mL and 1.96 × 10^−9^ μg/mL, for MTH, and 2.17 × 10^−11^ μg/mL and 3.04 × 10^−9^ μg/mL for MPHT, which correspond to the AChE biosensors, with different coatings. The first two values are based on a gold nanoparticle/three-dimensional graphene film (AuNPs/rGO) [116], and the second pair of values is based on AuNPs, with a VNSWCNTs overlay [107]. All detections show quite low limits in contrast to a classical method, based on the micro-extraction of emulsion using a micro funnel filter, followed by CG [42] (8.00 × 10^−3^ μg/mL, for MTH, and 1.30 × 10^−4^ μg/mL for the MPHT method based on HPLC-DAD [23]).

It was found that for diazinon (DZ) the lowest LOD was 5.00 × 10^−4^ μg/mL using an AChE biosensor based on the modulated fluorescence [121] of UCNPs-Cu^2+^, in contrast to the method of MFEM-MS [42], which registered a value of 4.00 × 10^−03^ μg/mL. In the case of the herbicide DU, 6.99 × 10^−6^ μg/mL was the lowest value of LOD, obtained by means of the square wave voltammetry technique [52], whereas by the DPX-HPLC/DAD method [23], a value of 6.00 × 10^−5^ μg/mL was obtained, this being the only case where a similar LOD value was obtained for the aqueous matrices.

Interestingly, when the insecticide ChP was analyzed in the agricultural matrices, such as fruits and vegetables, by means of an optical biosensor based on an immunosensor, established in microcantilever [59] and GC-MS [34], no significant difference in terms of the lowest LOD value of pesticides was determined.

A similar result was shown by Fu and coworkers, comparing the detection of ChP using an AChE biosensor, based on the OMC-CS electrode, modified with CeO_2_-CS (6.00 × 10^−4^ μg/mL) from SPCE [96], and the classical method, such as UPLC-Orbitrap MS16. However, a major difference is observed, when it is compared with a GC-MS/MS [39] (1.40 × 10^−2^ μg/mL).

An AChE biosensor based on a modified glassy carbon electrode [116] was used to determine the pesticide, MTH. It was shown to have an LOD of 1.50 × 10^−8^ μg/mL, compared to the value found by the classical GC-MS/MS [41] method (2.95 × 10^−3^ μg/mL).

By means of piezoelectric biosensors based on QCM (LOD values of 2.30 × 10^−5^ μg/mL of carbaryl) and nano colorimetric biosensors based on Thiocholine and H_2_O_2_ [71,75] (4.02 × 10^−5^ μg/mL of carbaryl), a significant difference in contrast to GC-MS/MS [41] was observed (1.20 × 10^−2^ μg/mL).

For the herbicide GF, lower LOD values of 4.56 × 10^−5^ μg/mL and 1.00 × 10^−5^ μg/mL are obtained. These are obtained by means of a magnetic nanoparticle fluorescence immunosensor [65] and a colorimetric immunoassay [64], respectively. The classical methods obtain a lower value of 2.50 × 10^−4^ μg/mL by HPLC-MS [43]. The fungicide IZ has an LOD value of 6.00 × 10^−4^ μg/mL, obtained by a white light reflectance spectroscopy immunosensor (WLRS) [60], which is low compared to a method based on HPLC-MS [41], which has an LOD value of 1.35 × 10^−2^ μg/mL.

In the specific case of the OP (chlorpyrifos) in water samples, liquid chromatography (LC) quadrupole-Orbitrap high-resolution tandem mass spectrometry (HRMS), reported by Casado et al., [123] and the optical methods (fluorescent) of Azab et al. [124] were compared. In this case, the LOD is 1 ng/L for the LC-HRMS, and for the optical method, the limit is 0.74 μmol/L. Therefore, it is concluded that the chromatography method has a greater sensitivity. The recovery for the chromatographic method was lower (51%), while that of the fluorescent method was 101.6%. As for the effect of the absolute matrix in LC-HRMS, it was 14% and a 5% for the optical method.

The inclusion of liquid chromatography (mass/mass) and new technologies has improved the results of inter-laboratory tests. In a study of pesticides in various matrices (cauliflower, leek, mandarin, pear, potato, pepper, broccoli, and spinach), carried out by Ferrer et al. in 2017 [29], satisfactory z-scores were detected in approximately 90% of samples. The best results were obtained for the liquid chromatography method, compared to gas chromatography [29]. In another study, 10 common fruits and vegetables (apple, banana, broccoli, celery, grape, green bean, orange, peach, potato and squash) that contain incurred and spiked pesticides were analyzed by low-pressure gas chromatography-tandem mass spectrometry (LPGC-MS/MS) and ultra-high-performance liquid chromatography (UHPLC)-MS/MS. They concluded that the preparation of the sample impacts more significantly the results obtained by any method evaluated. In addition, they evaluated the sample size, between 1–15 grams, and concluded that the bias increased by ≈10% when using 1–2 g portions vs. 15 g portions [125].

An additional study by Hildebrandt et al. evaluated the matrix using a portable biosensor prototype for the determination of neurotoxic pesticides (organophosphorus and carbamate) in water and food samples. They reported that differences from 13% to 73% were obtained, depending on the matrix. The higher the difference, the lower the matrix effect, as observed for seawater, which presented a mean difference of 73% between non-spiked and spiked samples, at 10 µg/L. This prototype works with a much lower amount of acetylcholinesterase (AChE) than in Ellman’s tests, and the quantity of the chlorpyrifos spiked in the sample was limited to concentrations of 5–10 µg/L [126].

Futuristic sensors for pesticides will be concentrated in order to ensure a low cost, portability, rapid detection, high sensitivity, and improvement of the reproducibility and real-time sensing capacity, with wireless networking using miniaturized designs [127]. The detection of pesticides can be converted into a quantifiable digital signal by hand-held devices, such as a smartphone, and then the detection results can be delivered to servers [128].

## 4. Conclusions

To summarize, sensors and biosensors for pesticide determination relating to agricultural activity have considerable potential to perform assays in a faster, simpler, low-cost and more sensitive manner, when compared with the traditional techniques for environmental control and the monitoring of various samples. In recent decades, there has been great interest in developing multiple types of biosensors. It is clear that enzymatic methods could play an essential role in the rapid in situ screening of large numbers of samples at a relatively low cost. Traditional chromatographic methods are used to accurately identify and quantify pesticides by the preliminary detection of samples. Therefore, the two approaches could complement one other. Nanomaterials improved the concept of flexibility, stability, optical transparency and compatibility, using microfabrication techniques, to high levels. They can be used to enhance the electrochemical response of the electro-active analytes and thus facilitate a multi-analysis and designation of portable on-site detection sensors. In spite of the massive development of biosensor instrumentation during the last two decades, the use of biosensors in the field is still restricted, as compared to medical applications. Additional attempts should be made to manufacture more reliable devices that will accelerate the detection of minor-scale pesticides in both laboratory and field conditions. For the large banana and pineapple industries in Costa Rica, it is necessary to rapidly determine pesticides, at the lowest cost and with a reasonable accuracy, for the protection of mankind. Nevertheless, there is a need to expand biosensor research to allows for a balance between agricultural activities and environmental preservation.

## Figures and Tables

**Table 1 molecules-24-02659-t001:** Quantities applied (kg/Ha) of banana pesticides by biocidal action, active ingredient and total (Atlantic Zone, Costa Rica).

Fungicides		Nematicides		Insecticides		Herbicides	
Ingredient active	kg per hectare	Ingredient active	kg per hectare	Ingredientactive	kg per hectare	Ingredient active	kg per hectare
Mancozeb	26.1	Terbufos	4.18	Bifenthrin	1.08	Glyphosate	2.34
Tridemorf	4.22	Fenamiphos	2.32	Chlorpyrifos	0.69	Paraquat	0.10
Chlorothalonil	1.14	Carbofuran	2.02			Diuron	0.04
Pyrimethanil	0.60	Ethoprophos	1.38			Diquat	0.004
Spiroxamine	0.52	Cadusafos	0.97			Glufosinate	0.004
Difenoconazole	0.37	Oxamil	0.34				
Piraclostrobin	0.19						
Azoxystrobin	0.19						
Bitertanol	0.18						
Tebuconzole	0.08						
Imazalil	0.08						
Thiabendazole	0.07						
Trifloxystobina	0.03						
Propiconazole	0.02						

**Table 2 molecules-24-02659-t002:** Comparison of the GC and HPLC methods for the determination of pesticides.

Pesticide Detected	Class of Pesticides	Detection Method	Limit of Detection	Sample Detected	References
Carbofuran	Nematicide	Liquid−liquid extraction & LC/ESI-MS	2 ng/mL	Human serum	[33]
		TLC-HPLC/DAD	6200 ng/mL	Blood	[33]
		GC-MS	0.1 ng/mL	Yam	[34]
		DPX-HPLC/DAD	0.02 µg/L		[23]
Chlorpyrifos	Insecticide	SPE-HPLC-MS/MS	0.05–0.5 ng/L	Water	[35]
		UHPLC-Orbitrap MS	0.06 μg/L	Frozen Fruit and Vegetables	[36]
		FPSE/GC-MS	0.088 ng/g	Vegetables	[37]
		GC-IMS	0.85 μg/L	Carbon sheets attached to magnetite (Fe_3_O_4_) nanoparticles	[38]
		GC−MS/MS	0.012 mg/Kg	High Water Fruits and Vegetables	[39]
		GC	0.01 mg/Kg	Apple	[40]
		HPLC-MS	0.01 mg/Kg	Pepper	[41]
		HPLC-MS	0.01 mg/Kg	Chili Sauce	[41]
		HPLC-MS	0.01 mg/Kg	Chili peppers	[41]
		MFEM-MS	6 ng/mL	Water	[42]
Glyphosate	Herbicide	HPLC-MS	0.25 μg/L	Environmental water	[43]
Terbufos	Insecticide	FPSE/GC-MS	0.033 ng/g	Vegetables	[37]

**Table 3 molecules-24-02659-t003:** Sensors developed for pesticides using various approaches.

Pesticide Detected	Class of Pesticides	Detection Method (Transducer)	Limit of Detection	Sample Detected	References
***Electrochemical***					
Carbofuran	Nematicide	Electrochemical screening assay: nano carbon black (CB)-based screen-printed sensor	8.0 × 10^−8^ mol/L	Grain: durum wheat, soft wheat and maize	[44]
		Cyclic voltammetry: Polyethylene terephthalate (PET)-derived activated carbon electrode material for non-enzymatics	0.03 µM	Agriculture	[45]
		Amperometric biosensors based on graphite-epoxy-AChE: electrode biocomposite 16% graphite	0.25 ppb	Water	[46]
		Electrochemical sensor based on molecularly imprinted polymer-reduced graphene oxide and gold nanoparticles	2.0 × 10^−8^ mol/L	vegetable	[47]
		Sensing interface of citrate-capped gold nanoparticles (AuNPs)/(3-mercaptopropyl)-trimethoxysilane (MPS)/gold electrode (Au)	1.0 nM	Fruits	[48]
Chlorpyrifos	Insecticide	Self-assembled monolayers (SAMs) of single-walled carbon nanotubes (SWCNT) wrapped with thiol-terminated single-strand oligonucleotide (ssDNA) on gold	1 × 10^−12 ^mol/L	Water	[49]
		Fluorine doped tin oxide (FTO)-based analytical sensor, coupled with highly conductive gold nanoparticles (AuNPs)	10 fM	Fruits and vegetables	[50]
Dimethyldithiocarbamate	Fungicide	Square wave voltammetry: enzyme-based biosensors	0.02 mg/kg	Plants	[51]
Diuron	Herbicide	Square wave voltammetry: Self-assembled films based on polypyrrole and carbon nanotubes composites	2.6 × 10^−7^ mol /L	Water	[52]
		A microbial biosensor based on the cyanobacterium *Anabaena variabilis*	0.003 nM	Water	[53]
Glyphosate	Herbicide	Immuno-Magnetic Assay with electrochemical sensors	5 ng/L	Beer	[54]
		Fiber-pencil graphite-modified electrochemical sensor	1.3 nM	soil & water	[55]
		MIP-AuNPs-CNTs	0.019 ng/mL	Soil	[56]
		Amperometry sensor based on Ni1−xAlx(OH)2NO3x·nH2O-layered double hydroxide (LDH)	0.9 mM	Water	[57]
		Potentiometric biosensor based on agarose-guar gum (A-G)-entrapped bio-nanoconjugate of urease, with gold nanoparticles (AUNps)	0.5 ppm	Water	[58]
***Optical***					
Carbofuran	Nematicide	Microcantilever-based immunosensor. Chemically modified by the crosslinking of L-cysteine (L-cys)/glutaraldehyde (GA)	0.1 ng/mL	Vegetables	[59]
Chlorpyrifos	Insecticide	Immunosensor white light reflectance spectroscopy (WLRS)-based Si (silicon) substrate	0,6 ng/mL	Water and wine grapes	[60]
Diuron	Herbicide	Diuron-molecularly Molecularly imprinted powers (MIPs)	0.013 μg/mL	Vegetables	[61]
		Microalgae-based optical bioassay	0.3 nM	Seawater	[62]
		Amperometry: Biosensor Toxicity Analyzer (BTA), consisting of the screen-printed sensor, Pt:Ag	1 nM	Water	[63]
Glyphosate	Herbicide	Colorimetric immunoassay: Using DNA-labeled gold nanoparticles	0.01–100 mg/L	Crops, vegetables, and fruits	[64]
		Immunosensor fluorescence magnetic nanoparticles	0.27 nM	Agricultural products	[65]
		Epic assay MC3T3-E1 cells using trypsin-EDTA	2.98 ± 0.18 nM	Agriculture	[66]
Imazalil	Fungicide	Immunosensor white light reflectance spectroscopy (WLRS)-based Si (silicon) substrate	0,6 ng/mL	Water and wine grapes	[60]
Tebuthiuron	Herbicide	Diuron-Molecularly imprinted powers (MIPs)	0.023 μg/ mL	Vegetables	[61]
Thiazophos	Insecticide	Immunosensor non-competitive SPR coated with a high density of carboxymethylated dextran	0.096 ng/mL	Cereals, vegetables, fruits	[67]
Thiabendazole	Fungicide	Immunosensor white light reflectance spectroscopy (WLRS)-based Si (silicon) substrate	0.8 ng/mL	Water and wine grapes	[60]
***Nano Colorimetric***					
Dimethyldithiocarbamate	Fungicide	Sodium dodecyl sulfate-capped silver nanoparticles (SDS-AgNPs)	9,1 ng/mL	Water	[68]
		Cetyltrimethyl ammonium bromide (CTAB)-capped copper nanoparticles (CTAB-Cups)	97.9 ng/mL	Tap water, tomato extract & mango juice	[69]
Chlorpyrifos	Insecticide	Citrate-capped gold nanoparticles (AuNPs)	118 ng/mL	Water & rice	[70]
		Thiocholine and H_2_O_2_	4.6 × 10^−8^ M	Apple juice & green tea	[71]
Carbaryl	Nematicide	Thiocholine and H_2_O_2_	2.3 × 10^−8 ^M	Apple juice & green tea	[71]
***Piezoelectric-Mass sensitive***					
Carbaryl	Nematicide	Quartz crystal microbalance (QCM) immunosensor, with acetylcholinesterase immobilized on one of the faces of the crystal	1 × 10^−7^ M	Water	[72]
		Quartz crystal microbalance (QCM) immunosensor: two-enzyme system (acetylcholine-esterase and choline oxidase)	1 mg/L	Water	[73]
		Quartz crystal microbalance (QCM) immunosensor: using monoclonal antibodies (MAbs)	11μg/L	Fruit juices	[74]
		Quartz crystal microbalance (QCM) immunosensor: based on a film bulk acoustic resonator, with antigens immobilized on the sensing surface of the resonator	2 × 10^−10^ M	Vegetables & crops	[75]
Carbofuran	Nematicide	Quartz crystal microbalance (QCM) immunosensor: acoustic micro-immunosensors immersed in an 11- mercaptoundecanoic acid and ethanolic solution	4.5 × 10^−6^ M	Grain	[76]
Chlorpyriphos	Insecticide	Quartz-crystal microbalance: the inhibitor, benzoylecgonine-1,8-diamino-3,4-dioxaoctane (BZE-DADOO), was immobilized on a monolayer of 11-mercaptomonoundecanoic acid (MUA), which was self-assembled on the gold surface of the sensor	1 × 10^−10^ M	River Water	[77]
		Quartz crystal microbalance (QCM) immunosensor, with molecularly imprinted polymers (MIPs) in gold nanoparticles	250 μg/L	Water	[24]
Parathion	Insecticide	Quartz-crystal microbalance: Photonic Immobilization Technique (PIT), functionalized with UV-activated antibodies	4 μg/L	Water	[78]
Glyphosate	Herbicide	Quartz crystal microbalance (QCM) immunosensor, with molecularly imprinted polymers (MIPs) in gold nanoparticles	250 μg/L	Water	[24]
***Chemiluminescence***					
Carbofuran	Nematicide	ECL energy transfer (ECRET) and carbon dot (C-dot)-tagged aptamers, as the recognition element	8.8 × 10^13^ mol/L	Vegetables	[79]
		CL Sensor Array Based on the LumAgNP−H_2_O_2 _System	24 μg/mL	Water	[80]
Glyphosate	Herbicide	Late-model and sensitive electrochemiluminescence (ECL): of ZnS quantum dots (QDs) on ordered mesoporous carbon (OMC) substrates	8.929 nM	Vegetable juice	[81]
Chlorpyriphos	Insecticide	CL Sensor Array Based on the LumAgNP−H_2_O_2 _System	24 μg/mL	Water	[80]
		Graphitic carbon nitride/bismuth ferrite nanocomposites (g-C_3_N_4_/BiFeO_3 _NCs)	0.033 ng/mL	Water	[82]
		Bimetallic Pt-Au nanoparticles were electrodeposited on a multi-walled carbon nanotube (MWNT)-modified glass carbon electrode (GCE)	0.08 nmol /L	Cabbage	[83]
Carbaryl	Nematicide	CL Sensor Array Based on the LumAgNP−H_2_O_2 _System	24 μg/mL	Water	[80]
***Fluorescence***					
Carbofuran	Nematicide	Carbon quantum dot (CQD)-based ratiometric nanosensor, exhibiting dual-emission fluorescence, coated with vitamin B12	12.2 μM	Soy sauce	[84]
Dimethyldithiocarbamate	Fungicide	Fluorescence of CdSe/ZnS quantum dots (QDs), capped with 3-mercaptopropionic acid	2 nM	Tomato and rice	[85]
Difenoconazole	Fungicide	Quantum dots (QDs) with Molecularly-imprinted polymers (MIP)	0.5 ng/mL	Apple, orange, and tomato juice	[86]
Diuron	Herbicide	The A-chlorophyll fluorescence of microalgae is assessed using *Chlorella vulgaris*, *Pseudokirchneriella subcapitata*, and *Chlamydomonas reinhardtii*	1 mg/L	Water	[87]

**Table 4 molecules-24-02659-t004:** Acetylcholinesterase biosensor developed for pesticides in fruits and in the environment.

Pesticide Detected	Class of Pesticides	Detection Method	Limit of Detection	Sample Detected	References
Carbofuran	Nematicide	uNCs-MnO_2_-AChE−CH	0.125 μg/L	Water	[105]
Chlorpyrifos	Insecticide	AChE/ATO/OMC/SPE	0.01 μg/L	Oilseed rape, Lettuce, Chinese cabbage	[32]
		NF/CS-AChE/Co–2Ni–B/GCE	2.83 pM	Water	[106]
		AChE/AuNPs/VNSWCNTs/Au	2.06 × 10^−6^ μg/L	Cabbage water, tap water, purified water, river water and lake water	[107]
		A sensitive electrochemical acetylcholinesterase (AChE) biosensor was developed on a polyaniline (PANI) and multi-walled carbon nanotube (MWCNT) core–shell-modified glassy carbon electrode (GC)	1.4 μmol/L	Apple, broccoli and cabbage	[4]
		Quartz-crystal microbalance: by the chemisorption of the AChE, thiolated with a heterobifunctional cross-linker	1.30 × 10^−9^ mol/L	Water	[99]
		AChE/OMC-CS/CeO_2_- CS/SPCE	0.01 μg/L	Oilseed rape, Lettuce, Chinese cabbage and Agaricus bisporus	[96]
		Enzyme electrode: AChE/MWCNT-SnO_2_- CHIT/SPE	0.05 μg/L	Cabbage apples tomatoes lettuce	[101]
		Enzyme electrode: AChE/MWNT-SnO_2_- CHIT/Au	2 μg/L	Lettuce leeks, pak choi	[101]
		Amperometric (CV): Nafion/ACh E/MWNTsSnO_2_- CS/Au	100 ng/L	Fresh lettuce, cucumber & pachouli	[102]
		Macromolecular polymer and carboxyl multi-wall carbon nanotubes (MWNTsCOOH), coated with acetylcholinesterases (AChE)	51.40 ng/L	Water	[103]
		AChE/Fe_3_O_4_/GR/SPE	0.02 µg/L	Vegetables	[108]
		AChE/MWCNTs/DCHP/SPE	0.05μg/L	Vegetables	[109]
		AChE/ZrO_2_/RGO	10^−1^ mol/L	Water	[110]
		AChE/MWCNTs/IL/SPE	0.05 μg/L	Vegetables	[111]
		AChE-Pin5COOH/Fe_3_O_4_ NP-modified GCE	9.1 nmol/L	Water	[112]
Malathion	Insecticide	AChE-CS/3DG-CuO NFs	0.31ppt	Water	[113]
		AChE/AuNPs/VNSWCNTs/Au	1.96 × 10^−6^ μg/L	Cabbage water, tap water, purified water, river water and lake water	[107]
		AChE/HCS@PAN	0.16 ng/mL	Fresh fruit and vegetables (apple, tomato, cucumber)	[114]
		Acetylcholinesterase biosensor, based on a glassy carbon electrode, modified with carbon black and pillar [5]	15 pmol/L	Wine, grape, and peanut	[115]
		AChE/Nafion/AuNPs/rGO/GCE	8.4 x10^−14^ mol/L	Tap water, mineral water and Chinese cabbage	[116]
		AChE/CS/Fe_3_O	0.3 nmol/L	Tomato and pond water	[117]
		NA/Ag@rGO-NH_2_/AChE/GCE	4.5 ng/mL	Water	[118]
		AChE-Pin5COOH/Fe_3_O_4_ NP-modified GCE	6.6 nmol/L	Water	[112]
Methyl parathion	Insecticide	AChE/AuNPs/VNSWCNTs/Au	3.04 × 10^−6^ μg/L	Cabbage water, tap water, purified water, river water and lake water	[107]
		AChE/Nafion/AuNPs/rGO/GCE	8.24 x10^−14^ mol/L	Tap water, mineral water and Chinese cabbage	[116]
		Origami paper-based electrochemical biosensor	2 μg/L	Water	[119]
Carbaryl	Nematicide	GC/rGO/AChE	1.0 nmol/L	Tomato	[120]
		Quartz crystal microbalance (QCM) immunosensor: two-enzyme system (acetylcholine-esterase and choline oxidase)	1 mg/L	Water	[73]
Diazinon	Insecticide	FL-AChE-ATCh-UCNPs-Cu^+2^	0.05 ng/mL	Water	[121]
Diuron	Herbicide	rGO–AuNP/SPE	10.0l g /mL	Lake and sea water	[122]

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
