# Peer review of "What are the Main Sensor Methods for Quantifying Pesticides in Agricultural Activities? A Review"

_molecules, 2019, doi:10.3390/molecules24142659_

Round 1

Reviewer 1 Report

Authors should stress novelty of the manuscript in comparison to the published papers / chapters / books.

What is the sensitivity, specificity of detection techniques described by the authors in comparison to other methods already described?
What is the application value and practical aspect of the detection techniques described in the publication?

The authors should also specify what is the repeatability of the methods used. Have inter-laboratory tests been carried out?

When planning experiments, analysts must make the right choice of the appropriate analytical (chromatographic) method. In the first instance, the analyst selects a method or a technique, which could serve the purpose he has in mind, that is, to determine a given analyte in a given, complex and multicomponent matrix.

Did Authors during comparing LOD for different pesticides, considered in what material (pesticide plus base) measurements were made, ie did they take into account the influence of sample material (matrices) on LOD for a given measurement method?

If possible, Authors should add evaluate of  influence of matrix.

Liquid chromatography–mass spectrometry (LC–MS) uses directed sample preparation towards generic methods that are able to extract as many residues as possible, as its selectivity avoids the use of extensive clean-up procedures, despite the complex composition of the matrix. The introduction of high-resolution mass spectrometry (HRMS) analyzers, such as time-of-flight (TOF) and Orbitrap, has allowed the development of non-target screening and unknown identification schemes, which can run independently or can be combined with the target analysis.

Various mass analyzers offering these possibilities are already used routinely in combination with gas or liquid chromatography, among which triple quadrupoles (QqQs) and ion traps (ITDs) are in common use. More recent technologies are linear ion trap (LITs), orbital trap (OrbitrapTM) and new-generation of hybrid instruments, for example, quadrupole time-of-flight (QqTOF), quadrupole–linear trap (Qq–LITs), or linear–orbital traps (LTQ–OrbitrapTM), which are gaining widespread acceptance in several application areas. All these recent instruments offer advantages such as high scanning speeds, accurate mass measurement (QqTOF, LTQ–OrbitrapTM) and increased sensitivity (LITs and new generation of QqQs). The application range of multi-dimensional MS is today extremely wide, both in terms of target compounds and in terms of different possible acquisition modes. This last capability confers not only very sensitive and specific quantitative target measurements, but also powerful untargeted ‘fishing’ approaches based on the detection of typical product/precursor ions or neutral species belonging to a class of substances.

Other detection techniques can also be used, e.g. new LED UV-absorption detector.

Also, antibody-based assays such as enzyme-linked immunosorbent assay (ELISA) and colloidal gold immunoassay (CGIA), as well as also fluorescent microsphere immunoassays (FMIAs), can be used for the rapid detection of some of xenobiotic residues.

Miniaturization in chromatography has emerged with capillary liquid chromatography (cLC) and nano-liquid chromatography (nano-LC) as alternatives to HPLC.

Some references are also omitted (please see example):

https://www.crcpress.com/High-Performance-Liquid-Chromatography-in-Pesticide-Residue-Analysis/Tuzimski-Sherma/p/book/9781466568815

https://www.crcpress.com/Determination-of-Target-Xenobiotics-and-Unknown-Compound-Residues-in-Food/Tuzimski-Sherma/p/book/9781498780131

Reviewer 2 Report

The article deals with methods of detecting trace quantities of pesticides at the molecular level and as such corresponds to the profile of the journal. The subject matter connected with quality (purity) of food, water and soill is important, but the article itself is a bit chaotic, which is why I propose its major revision.

1. The authors did not take into account the important optical spectroscopic method of detection and analysis at the molecular level of trace amounts of substances, which is Raman spectroscopy, in particular its SERS variant. The example is: Liu, B., Zhou, P., Liu, X. et al. Food Bioprocess Technol (2013) 6: 710. https://doi.org/10.1007/s11947-011-0774-5.

2. The article is a bit chaotic, the authors orient it to the study of the level of contamination of banana and pineapple crops, but they should clearly determine whether they mean contamination of the fruits, soil and / or water themselves.

3. The authors should clearly indicate to what extent it is necessary to determine the contamination of the fruits themselves (on the surface or in the pulp?), soil and / or water. Not just the amount of pesticides used per hectare.

4. In chapter 2, the authors exchange methods / groups of methods. Part 2.3 Cyclic voltametry is too specific in relation to the others (instrument model, parameters), while the part 2.4 Optical method is too general - there are not even all the methods listed. Details from part 2.3 are more in line with the scope of Chapter 3.

5. Authors should recalculate and provide a comparable limit of detection (the same unit) in Tables 3 and 4.

6. The description of the detection method (transducers) in Table 3 should be better standardized: electrode materials in all electrochemical solutions, measurement methods for all optical solutions, etc.

7. The Authors describe studies of various crop-related materials. They should discuss how much these materials are in clear-cut close-up of bananas and pineapples and the measurement problems associated with them.

8. The Authors describe a significant number of measurement experiments, both in tables and in the descriptive part of Chapter 3. Due to the size of the study, it is difficult to combine items from the table with equivalents in the text. Maybe it is worth introducing own markings of each solution and apply them both in the text and in the tables. There is no need to repeat in the text data already contained in the tables.

9. When describing electrochemical sensors, the authors should also discuss the possibility of repeated use of electrode data (the problem of electrode cleaning).

10. The authors should also refer to diamagnetic electrodes (for example doped with boron BDD) and metallic (eg platinum) electrodes.

11.It is advisable to unify the description of the research of other teams. Providing only selected names of authors of presented research reduces the readability of the article.

12. The authors should clearly define their goal (the methodology of future research on bananas and pineapples) - chapter 4 - do they seek the best (group of the best) universal methods, i.e. they want to build a mobile measurement system for plantation control or maybe they are looking for the highest method for each pesticide separately?

13. Did Authors during comparing LOD for different pesticides,  considered in what material (pesticide plus base) measurements were made, ie did they take into account the influence of sample material on LOD for a given measurement method? 

14. Did the Authors analyze the possible cost of measurement, mobility of the apparatus and the time of analysis to equalize individual methods. Did the authors analyze the possibility of detecting pesticides based on laboratory tests or field tests?

15. Did the authors develop the concept of a measuring instrument based on a literature analysis? Did the study of baboons and pineapples beą require special arrangements or maybe you can use similar systems for examining eg cabbage or grain?

16. Each of the measurement methods (not just groups) should be described (principle of operation) though one sentence.

Round 2

Reviewer 2 Report

The article corresponds well to the subject of the journal. The subject is up to date. A full review and analysis of methods for detecting trace amounts of pesticides is presented. Corrections recommended in review No. 1 have been introduced. I recommend the article for printing in the review articles category.